# Anthropomorphic Design and Self-Reported Behavioral Trust: The Case of a Virtual Assistant in a Highly Automated Car

Clarisse Lawson-Guidigbe [1,2], Kahina Amokrane-Ferka [1,*] , Nicolas Louveton [3], Benoit Leblanc [2], Virgil Rousseaux [1] and Jean-Marc André [2]

1   IRT SystemX, 91120 Palaiseau, France
2   Laboratoire IMS CNRS UMR 5218, Bordeaux INP-ENSC, Université de Bordeaux, 33400 Talence, France
3   CeRCA CNRS UMR 7295, Université de Poitiers, Université François-Rabelais de Tours, 86073 Poitiers, France
*   Correspondence: kahina.amokrane-ferka@irt-systemx.fr

**Abstract:** The latest advances in car automation present new challenges in vehicle–driver interactions. Indeed, acceptance and adoption of high levels of automation (when full control of the driving task is given to the automated system) are conditioned by human factors such as user trust. In this work, we study the impact of anthropomorphic design on user trust in the context of a highly automated car. A virtual assistant was designed using two levels of anthropomorphic design: "voice-only" and "voice with visual appearance". The visual appearance was a three-dimensional model, integrated as a hologram in the cockpit of a driving simulator. In a driving simulator study, we compared the three interfaces: two versions of the virtual assistant interface and the baseline interface with no anthropomorphic attributes. We measured trust versus perceived anthropomorphism. We also studied the evolution of trust throughout a range of driving scenarios. We finally analyzed participants' reaction time to takeover request events. We found a significant correlation between perceived anthropomorphism and trust. However, the three interfaces tested did not significantly differentiate in terms of perceived anthropomorphism while trust converged over time across all our measurements. Finally, we found that the anthropomorphic assistant positively impacts reaction time for one takeover request scenario. We discuss methodological issues and implication for design and further research.

**Keywords:** human–machine interface; virtual assistant; trust; anthropomorphism; test methods; user studies; evaluation; autonomous car



## 1. Introduction

During the last four decades, the automotive industry has introduced important innovations in the automation of driving tasks. These are often driver assistance systems which, depending on their level, take over some driving tasks or all of them. The Society of Automotive Engineers (SAE) proposes a six-level description of driving automation ranging from fully manual (L0) to fully automated (L5; see Appendix B) driving. The highest levels of automation (L3 or higher), although not yet available to the public, present significant challenges for the acceptance and adoption of this technology. Indeed, despite the benefits of automated vehicles and predictions of the rate at which they will be adopted, barriers remain [1,2]. These include the possible consequences of automation on the driver's mental workload [3] and situational awareness [4,5] but mainly on the lack of user trust [6].

Trust has been widely discussed in the literature and has different definitions depending on the authors. It can be considered as a psychological state [7], a behavioral intention [8], or a behavior [9]. In this paper, we rely on the definition of trust proposed in [10]: "a psychological state of expectation resulting from knowledge and evaluations related to the operator's trust referent in a specific context and guiding his/her decision to use automatic control". Trust in automated systems is measured using different tools across

studies. In [11,12] the authors propose a detailed literature review of trust measurement tools which include self-reported scales, behavioral measures, and physiological measures. The most widely used self-reported scale is from [13]. This scale includes a wide range of factors (dependability, reliability, confidence familiarity, etc.) and has been used for a wide range of automated systems. Other scales are also developed to suit specific systems such as autonomous vehicles [14,15] or robotics [16]. Behavioral measures of trust involve observation and recording of participants' behavior during interaction with the system and interpretation of these behaviors as a trust indicator. For example, a higher trust level in the automated system is equated with an attenuated startle response in a risky situation [14], less monitoring of the system (reliance behavior) [17], or the withdrawing of the participant's own decision to comply with automation decision (compliance behavior) [18]. Similarly, physiological measures such as lower heart rate [14], gaze behavior [19], or electrodermal response [20] are used to indirectly assess participants' trust level.

Trust, and more specifically under-trust, has been identified as one of the most important challenges to the adoption of automated vehicles [6]. Under-trust presents a risk of rejection of driving automation and, therefore, threatens its large-scale adoption and the possibility of societal benefit from its advantages. Under-trust is the opposite of over-trust, which itself presents other challenges, e.g., overdependence on automation and safety risks in situations where the automated system does not function as it should [21].

According to some authors, well-calibrated trust would facilitate the acceptance of automated systems [22] and reduce the risks of misuse and non-use (disuse) [23]. For automated vehicles, some studies show that the solution to under-trust is integrating anthropomorphic design into the vehicle's human–machine interface [24,25].

Anthropomorphism represents, according to various authors, the tendency for humans to attribute human abilities and characteristics to machines or inanimate objects, such as personality, feelings, rational thinking [26,27], or intentions [28]. Consequently, machines are treated as entities (Media Equation Theory) capable of engaging in social interactions [26]. Some authors [29] explain that assigning human characteristics to a machine would make it more familiar, more explainable, and more predictable. Anthropomorphic design, whose goal is to elicit an anthropomorphic perception in the user, simulates life in inanimate objects through design [30,31]. In robotics, three components of anthropomorphic design are identified in [32]: (1) The form of the robot; (2) Its behavior; (3) Its interaction and communication with humans. Anthropomorphic design in automated vehicle interfaces [33,34] makes use of different social stimuli such as visual appearance, voice and natural language, and non-verbal behaviors.

This work focused on a specific type of virtual assistant interface. Our goal was to study the influence of anthropomorphic attributes (voice, natural language, and visual appearance), alone or combined, on the perception of anthropomorphism and trust in the automated system. This led us to the following research questions:

(Q1): Does increasing anthropomorphic attributes lead to an increase in the perception of anthropomorphism?

(Q2): Does increasing anthropomorphic attributes lead to an increase in the level of trust in the automated system?

(Q3): Does the perception of anthropomorphism correlate with trust?

(Q4): Does increasing anthropomorphic attributes lead to better driving performance?

To answer these questions, three user interfaces with different levels of anthropomorphism were designed and tested: baseline, vocal (natural language), and visual assistant (natural language combined with visual appearance).

## 2. Materials and Methods

### 2.1. Driving Simulation and Automated Driving System (Environment)

The study was conducted in a fix-base driving simulator housing a cockpit. This cockpit was based on a Renault Espace designed to represent a vehicle equipped with two automation levels: manual level (L0) and autonomous level (L3). The autonomous level

can be activated on a highway or expressway and can replace the driver to speeds up to 110 km/h. This study more specifically focused on handover (transition from L0 to L3) and handback (transition from L3 to L0) situations. Different driving scenarios were used in the tests (Table 1). The driving simulation was generated using AVSimulation SCANeR Studio 1.9 software and projected on a 180° hemispheric screen to provide a driving experience as immersive and as ecologically valid (i.e., generalizable toward real-world application) as possible (Figure 1).

**Table 1.** The different scenarios.

| Scenario | Description | Related Research Question and Aims |
|---|---|---|
| Training phase | | |
| Training scenario 1 | This scenario allows the participant to discover the different user interfaces of the cockpit and the activation/deactivation procedures for the autonomous mode. | |
| Training scenario 2 | This scenario allows the participant to discover the hand back procedure (takeover requested by the system). In this situation, the driver has 60 s to take over the driving. | |
| Experimental phase | | |
| Activation Scenario | This scenario is the first of the experimental phase. First, the driver must merge onto a highway in manual mode before activating the autonomous mode in light traffic. It aims to evaluate how increasing anthropomorphic attributes in the interface increases participants' performance, specifically their ability to activate the autonomous mode. | Q1, Q2, Q3, and Q4. |
| Takeover request in 60 s (TOR 60)  | The takeover request in 60 s (TOR 60) scenario starts with 6 min of automated driving on the same road as in the activation scenario. It ends with a non-urgent takeover request during which the participant has 60 s to regain control of the vehicle. This request is triggered by the system and due to the presence of a highway exit on the path followed by the vehicle. | Q1, Q2, Q3, and Q4. It aims to evaluate how increasing anthropomorphic attributes in the interface impact participants' performance, specifically their ability to handle a non-urgent takeover request, and also the variation in their trust and perception of anthropomorphism. |
| Takeover request in 10 s (TOR 10)  | The takeover request in 10 s (TOR 10) scenario starts with 4 min of automated driving on a straight highway in moderately dense traffic. It ends with an urgent takeover request during which the participant has only 10 s to regain control of the vehicle in a complex situation (dense traffic, with a vehicle stopped in the participant's vehicle lane vehicle lane and another in the left blind spot). This request is triggered by the system and due to a sensor failure. | Q1, Q2, Q3, and Q4. It aims to evaluate how increasing anthropomorphic attributes in the interface impact participants' performance, specifically their ability to handle an urgent takeover request, and also the variation in their trust and perception of anthropomorphism. |

The simulator is also used to test multimodal human–machine interface (HMI) configurations. To do so, the cockpit is equipped with screens (cluster, central display, head-up display, steering wheel display, and mirrors), LEDs (side windows, windshield, and steering wheel), speakers (cockpit and headrest), and seat actuators. Haptic feedback in the steering wheel can also be controlled using a Sensodrive motor.

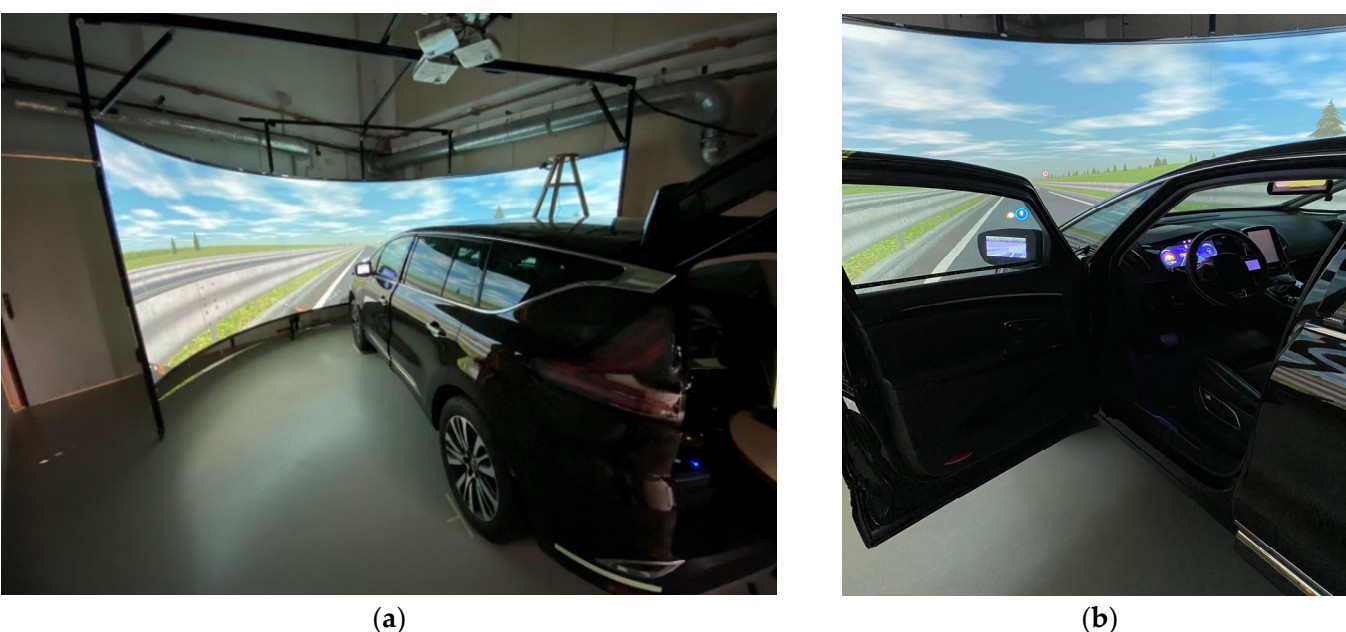

(**a**)                                                              (**b**)

**Figure 1.** (**a**) Simulation environment and cockpit, (**b**) view of the interior of the cockpit.

*2.2. Experimantal Factors*

Three interfaces were compared: baseline (Figure 2); vocal assistant, the interface integrating the voice of the assistant (Figure 3); and visual assistant, an interface integrating a combination of a visual representation and the voice of the assistant (Figure 4). Each group of participants tested only one of the three interfaces.

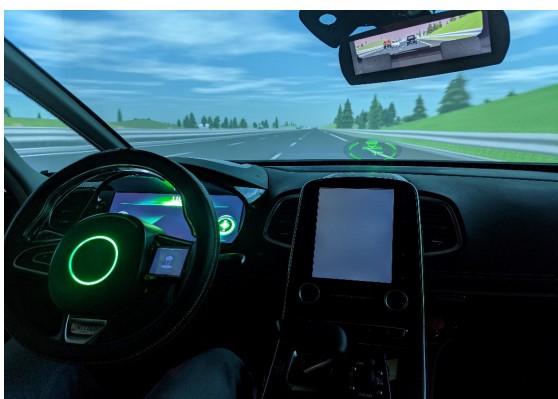

**Figure 2.** Baseline interface.

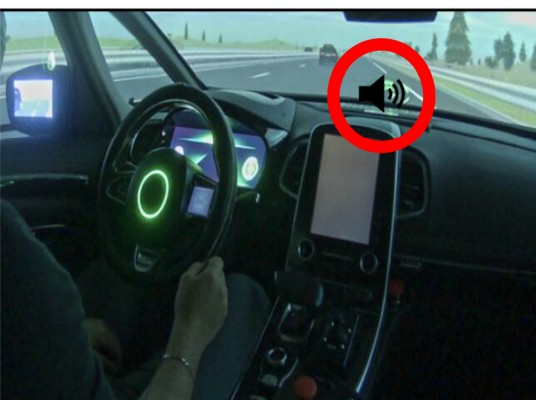

**Figure 3.** Baseline interface enriched with vocal assistant.

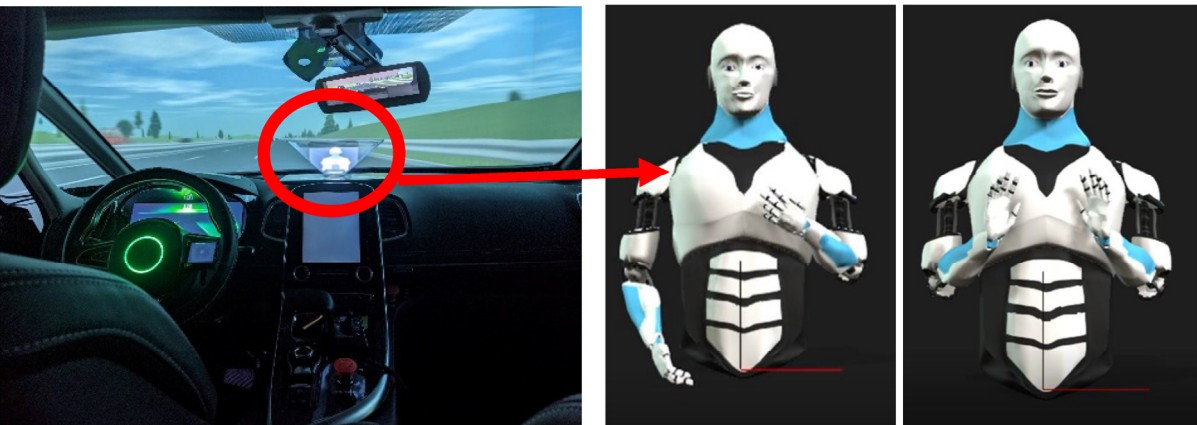

**Figure 4.** Baseline interface base and vocal assistant enriched with the visual representation.

### 2.2.1. Baseline Interface

This is a multimodal interface (Figure 2) that integrates the following:

1.  Visual interface including, on the dashboard, information on speed, driving mode availability, and time available in automated mode. On the steering wheel, a small integrated screen allows the driver to change driving modes (automated to/from manual) and displays the status of the autonomous mode (available, activated, and deactivated). The head-up display (HUD) replicates most of the dashboard's information. In the middle of the windshield, an icon replicates the status of the autonomous mode and time remaining before having to take over the driving.
2.  Sound interface comprising alert tones with different tonality and rhythms played locally in the driver's headrest.
3.  Haptic messages are created through the integration of actuators in the backrest and seat of the driver.

### 2.2.2. The "Vocal Assistant" Interface

The vocal assistant corresponds to the baseline condition enriched by vocal messages emitted by the assistant. These messages are complete sentences that include actions that the driver must perform and the associated explanations (see examples in Table 2). They are pre-recorded using "Any Text to Voice" software (version 3.6). The vocal messages are spatialized, emitted from a tablet in the middle of the dashboard which integrates the assistant. In this condition, the assistant uses the same iconic visual representation in the windshield as in the "baseline" interface (Figure 2).

**Table 2.** Example of vocal assistant messages.

| Situation | Message Example | Message Type |
|---|---|---|
| AD activated | You can regain control at any time. This is only possible if your hands are on the steering wheel and your eyes are on the road. I wish you an excellent journey. | Information |
| TOR 60 at 60 s before takeover | You must take over control to take the next exit. I won't be able to drive in 60 s. | Explanation |
| TOR 60 at 10 s before takeover | Take over control. | Explanation |
| TOR 10 | Sensor failure detected; vehicle stops; take over control! | Explanation |

### 2.2.3. The "Visual Assistant" Interface

The visual assistant interface combines baseline and vocal assistant interfaces with a humanoid visual representation [25]. This representation is projected in three dimensions using a tablet and a pyramidal structure that generates a hologram that appears in the middle of the dashboard when autonomous mode is activated (Figure 4).

### 2.3. Participants

Thirty-six people participated in the experiment. Following a between-subjects experimental design, three groups of twelve were formed and each group tested only one of the three interface conditions. Participants were selected based on the following criteria: minimum 3 years of driving experience, regular driving activity (at least 3 times a week) on several types of infrastructures (city, highway, and expressway), and minimal knowledge on driving assistance systems. In addition, participants had to answer a question regarding their propensity to use automation technology. Participants who showed no interest in the technology were rejected. Other exclusion criteria, such as motion sickness or simulator sickness as well as visual and/or hearing problems were applied. In the three groups, a gender and age balance were respected.

### 2.4. Procedure

Individual test sessions lasted 2 to 3 h and were divided into five steps (see Figure 5). The first step or "welcome session" of participants contained a detailed description of the goal of the experiment and the data that were collected. The second step consisted of training participants to use the driving simulator through a computer tutorial whose aim was to explain how the automated driving mode works (including activation and deactivation procedures), the vehicle user interfaces, and the operating limits of the autonomous mode. The tutorial was followed by 2 training scenarios (as shown in Table 1) and finally a trust survey measuring global and multidimensional trust [10]. The third step was the experimental phase during which participants had to complete four driving scenarios (see Table 2). After each scenario, participants completed an elicitation interview [35] and a trust survey. The trust survey at this stage was measuring global and multidimensional trust (adapted from [36]; see (Table 3) and Appendix A. In the fourth step, at the end of the experiment, participants completed the perceived anthropomorphism survey [14] and the global trust survey [37]. The fifth and final step was an informal discussion to collect participants' feelings and thank them.

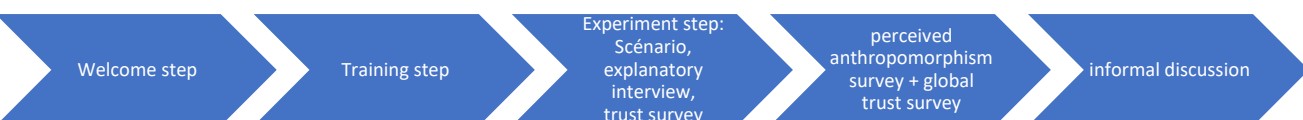

**Figure 5.** Test session procedure.

### 2.5. Tasks

To avoid bias as much as possible, the following instructions were read to the participant before each driving scenario of the experimental phase:

> "The driving scenario starts at on a highway ramp. I ask you to start the vehicle and merge onto the highway. Once on the highway, activate the autonomous mode as soon as it is proposed. When you have completed the activation, I will give you a smartphone that you will use to play the 2048 game. If the takeover alert is triggered at any time during the scenario, you must stop the game and regain driving control as soon as possible".

**Table 3.** Trust scales used before, during, and after the experiment.

| Trust Scale Used before Experiment [10] (Initial and Post-Training) | Trust Scale Used During Experiment [36] (after Each Driving Scenario) | Trust Scale Used at the End of Experiment [37] |
|---|---|---|
| 1—Do you think you know the automated car? 2—How much trust would you have in the automated car? 3—Do you think the automated car is useful? 4—Do you think the automated car is necessary? 5—Do you think that the automated car would interfere with your usual driving? 6—In everyday life, you tend to take risks 7—You tend to trust people 8—You believe that trusting someone means being able to trust them with something to do 9—You believe that it is necessary to know a person well to trust him/her 10—You think that the decision to trust someone depends on how you interact with them 11—You are generally suspicious of new technologies (cell phones, computers, internet, microwave ovens, etc.) 12—You think that new technologies are interesting 13—You think that new technologies are dangerous **What risks would you associate with using an automated car?** 14—The risk of driving more dangerously 15—The risk of not knowing how to use the system 16—The risk of an accident with the car ahead 17—The risk of an accident with the car behind 18—The risk of being dependent on the system 19—The risk of losing the pleasure of driving **What benefits would you get from using an automated car?** 20—Less stressful driving 21—Lightening the driving task (the automated car would allow me to perform a secondary activity such as reading or using my phone) 22—Easier driving task 23—Improved driving comfort 24—Safer driving | 1—I trust the automated car in this situation 2—I would have performed better than the automated vehicle in this situation 3—In this situation, the automated vehicle performs well enough for me to engage in other activities (using my smartphone) 4—The situation was risky 5—The automated vehicle made an unsafe judgement in this situation 6—The automated vehicle reacted appropriately to the environment 7—In this situation, the behavior of the automated car surprised me | Overall, how much do you trust the automated car? |

### 2.6. Data Collection and Analysis

Trust is a complex factor to assess: the tools that are used, their number (one or more), and the moment when trust is measured (before, during, or after interaction) varies across studies and may have an impact [38,39] on results. The variety of measurements in the scientific literature can be sub-divided into three categories: self-reported scales, behavioral measures, and physiological measures [11,12].

In this study, trust was measured using the model proposed in [40]. Based on our literature review, this model best matched the measurement of trust in the context of interaction with an automated vehicle at the time of our experimentation. Indeed, this model proposes various factors that would influence trust before and after interaction with the automated system.

Three different scales were selected to measure trust before, during, and after the experiment (Table 3). Different selection criteria were used to best assess the variability of trust dimensions among the three stages of interaction described in Hoff and Bashir's model [3]. However, in our results (Section 3) we distinguish global trust and multidimensional trust. On one side, global trust is measured using the same question across the three scales. This enables the evolution of trust to be measured from the beginning to the end of

the experiment. On the other side, multidimensional trust represents the dimensions of trust that differ before and during interaction with the system.

Initial and post-training trust was measured using the trust scale from [10]. As suggested by the model (Figures 6 and 7), this scale is made up of 24 questions measuring personality traits (questions 6 to 13) as well as pre-existing knowledge (questions 1 to 5 and 14 to 24). In the results section (Section 3), initial and post-training global trust refers to the answer to question 2 of this scale (*How much trust would you have in the automated car?*). Initial and post-training multidimensional trust scores are obtained by averaging the scores of all 24 questions (questions 5, 9, 13, 14, and 19 being negative; their scores were inverted before calculating the average).

During the experiment, trust was measured after each driving scenario using the scale taken from [36]. This scale was specifically designed as a first approach to empirically validate Hoff and Bashir's model [40]. Global trust is the answer to question 1 of this scale (*I trust the automated car in this situation*). The multidimensional trust score was obtained by averaging the scores of all 7 questions (questions 2, 4, 5, and 7 being negative; their scores were inverted before averaging—see Table 3).

Finally, trust was measured at the end of the experiment using the 1-item questionnaire from [37] (*Overall, how much trust do you have in the automated car?* See Table 3).

An initial and post-learning global trust score was computed based on item 2 (*How much trust would you have in the automated car?*) of the initial trust scale (adapted from [10]; see Appendix A). A global trust score was calculated for each driving scenario based on item 1 (*How well do you trust the automation in this situation?*) of the situational trust scale [36]. Finally, the post-experimental global trust was measured using a one-item survey [37] at the end of the experiment.

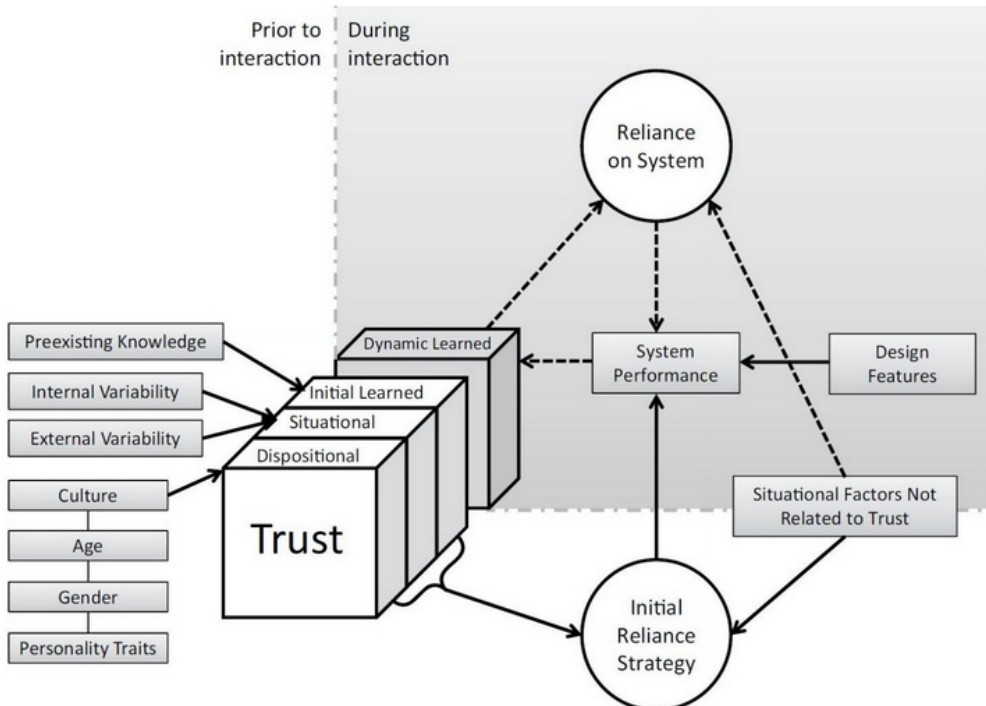

**Figure 6.** Factors that influence trust prior to interaction (initial trust in this paper) from Hoff and Bashir's model [40]. According to this model, prior to interaction with a system, user trust is influenced by his personality traits, age, gender, culture (dispositional trust), pre-existing knowledge of the technology (initial learned trust) which can be acquire through advertisement, peer recommendation or experience with similar system, and the situation in which the interaction is taking place (situational trust).

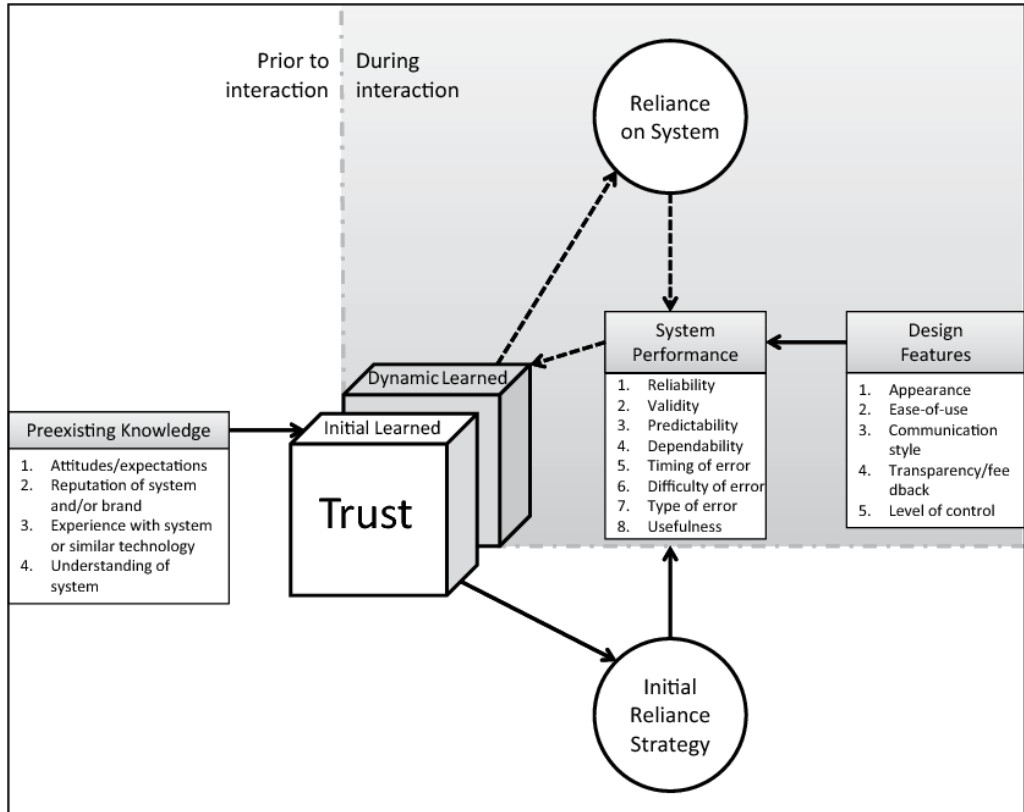

**Figure 7.** Factors that influence trust during interaction (measured after each scenario in this paper) from Hoff and Bashir's model [40]. During interaction, user trust is influenced by system design features and performance. The system performance is highly influenced by user reliance which is dynamically formed and adjusted during the interaction.

Perceived anthropomorphism is the degree to which users perceive humanness in a machine. In some studies, it has been measured through features that are uniquely human (agreeableness, openness, and civility) or typically human (extraversion, emotionality, warmth, and openness) [41,42]. Some studies explored sociability, human-likeness, and machine-likeness of robots [43,44]. Others explored the robot's consciousness, fakeness, and movement artificialness [45].

In this study, a self-reported scale from [14] was used to measure perceived anthropomorphism. It includes four questions which focused on the system's "mental" capacities. These questions asked participants how smart they thought the car was, how well the car could feel what was happening around it, how well it could anticipate what was about to happen, and how well it could plan a route. We chose this scale because we adopt the author's perspective on perceived anthropomorphism and because of the similarities between [14] and our experiment (an automated driving car with anthropomorphic features).

A perceived anthropomorphism score was calculated for each participant by averaging the answers to each of these four questions. The closer the score was to 10, the more the interface was perceived as anthropomorphic.

To test our hypotheses, we performed inferential tests. Quantitative data were analyzed using statistical tests adapted to the type of data. The statistical significance threshold has been set to the most common value in the field ($p$-value lesser than 0.05). These tests were combined with post hoc comparisons when further comparisons were needed (with $p$-value correction adjustments). Driving simulator data were quantitative (reaction time) while trust scale ratings (Likert scales) were combined to calculate a trust score. Statistical analyses have been performed on these scores.

### 3. Results

#### 3.1. Perciption of Anthropomorphism

The hypothesis relating to Q1 was that perceived anthropomorphism will be higher when the interface integrates the visual assistant (holographic representation) compared to the interface integrating the voice assistant or the baseline interface.

Analysis of perceived anthropomorphism scores (Figure 8) did not support the hypothesis. Each interface received a relatively high perceived anthropomorphism score, between 7.5 and 8.5 out of 10. The vocal assistant interface had the highest score (8.5) compared to the visual assistant and baseline conditions (7.5). These differences were not statistically significant as shown by the Kruskal–Wallis test (chi squared = 0.99, df = 2, $p$-value = 0.61)

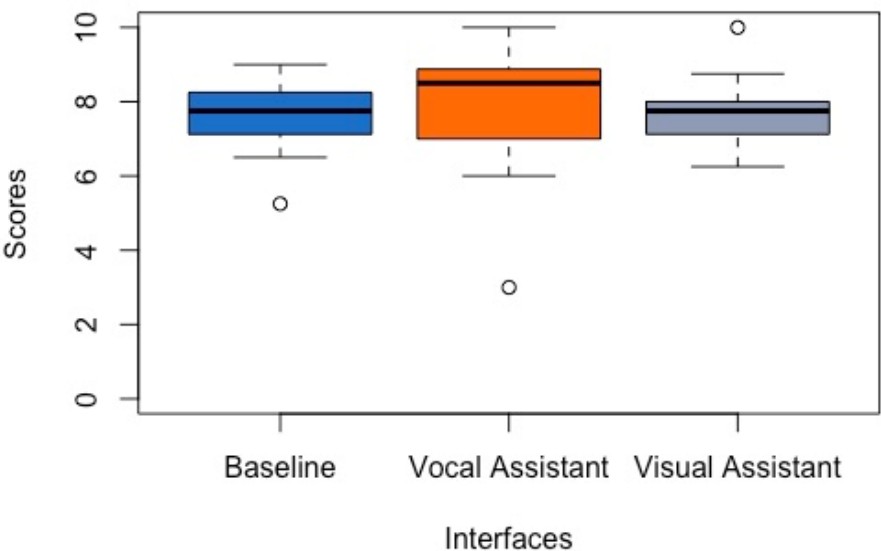

**Figure 8.** Perceived anthropomorphism score for each interface.

#### 3.2. Trust

These analyses answer research question Q2. Our hypothesis was that trust levels are higher for participants experiencing the visual assistant than for those who have the voice assistant or the baseline. Table 4 presents the global trust scores obtained at each stage of the experiment.

**Table 4.** Average global trust score (interquartile ranges in parentheses), the * sign indicates when the statistical test results are significant.

|  | Initial | Post-Training | Activation | TOR 60 | TOR 10 | Post-Experiment |
|---|---|---|---|---|---|---|
| Baseline | 7 (1) | 8 (0) | 10 (1.25) | 9.5 (1.25) | 7 (3.5) | 8 (2) |
| Vocal Assistant | 6 (1) | 7 (1.25) | 8 (2.25) | 9 (1.25) | 8 (4.25) | 7.5 (1.25) |
| Visual Assistant | 6 (1) | 7 (2) | 8.5 (2) | 9 (1.5) | 7 (1.5) | 7.5 (2) |
| Kruskal–Wallis test | 0.0002 * | 0.0284 * | 0.0603 | 0.415 | 0.7895 | 0.6246 |

Results for the three driving scenarios (Activation, TOR 60, and TOR 10) show no significant differences among the three interfaces. However, initial and post-training global trust (Tables 5 and 6) are significantly different between the baseline and vocal assistant and between the baseline and visual assistant interfaces.

**Table 5.** Initial global trust.

|  | Visual Assistant | Vocal Assistant |
|---|---|---|
| Vocal Assistant | 1.00 |  |
| Baseline | 0.0005 * | 0.0012 * |

(*p*-values from the Mann–Whitney pair test using a Bonferroni correction; alt. hypothesis = Baseline > Assistant Vocal and Visual), the * sign indicates when the statistical test results are significant.

**Table 6.** Post-training global trust.

|  | Visual Assistant | Vocal Assistant |
|---|---|---|
| Vocal Assistant | 1.00 |  |
| Baseline | 0.029 * | 0.031 * |

(*p*-values from the Mann–Whitney pair test using a Bonferroni correction; alt. hypothesis = Baseline > Assistant Vocal and Visual), the * sign indicates when the statistical test results are significant.

Multidimensional trust scores showed no significant difference between the three interfaces except for the post-training score where participants had more trust with the baseline interface than with the two anthropomorphic interfaces (vocal and visual; see Tables 4 and 7). A two-by-two comparison of each interface confirmed this result (Table 8) for both global (Table 4) and multidimensional trust (Table 8).

**Table 7.** Multidimensional trust (median scores and interquartile range), the * sign indicates when the statistical test results are significant.

|  | Initial | Post-Training | Activation | TOR60 | TOR10 |
|---|---|---|---|---|---|
| Baseline | 6.7 (1.67) | 8.03 (0.98) | 8.43 (1.14) | 9.07 (1.00) | 5.71 (3.64) |
| Vocal Assistant | 6.07 (1.68) | 6.78 (1.70) | 8.79 (1.64) | 9.29 (1.39) | 6.71 (2.96) |
| Visual Assistant | 5.74 (1.05) | 6.55 (1.54) | 8 (1.25) | 8.93 (1.11) | 5.29 (2.61) |
| Kruskal–Wallis test | 0.1271 | 0.0122 * | 0.3823 | 0.5545 | 0.7003 |

**Table 8.** Multidimensional post-training trust.

|  | Visual Assistant | Vocal Assistant |
|---|---|---|
| Vocal Assistant | 1.00 |  |
| Baseline | 0.0068 * | 0.0307 * |

(*p*-values from the Mann–Whitney pair test using a Bonferroni correction; alt. hypothesis = Baseline > Assistant Vocal and Visual), the * sign indicates when the statistical test results are significant.

Global and multidimensional trust scores were mostly equivalent, except for the initial trust measurement: multidimensional trust scores were not different across the three interface conditions (Table 5) as opposed to global trust score (Table 4). In summary, these results confirm our hypothesis only for the global trust score in the initial trust measurement.

*3.3. Correlation between Anthropomorphism and Trust*

The third research question (Q3; see Section 1) examined the relationship between perceived anthropomorphism and trust and hypothesized a correlation between them. A medium but significant correlation between perceived anthropomorphism and post-experimental global trust ($\rho = 0.45$, *p*-value < 0.05) supports this hypothesis (Figure 9).

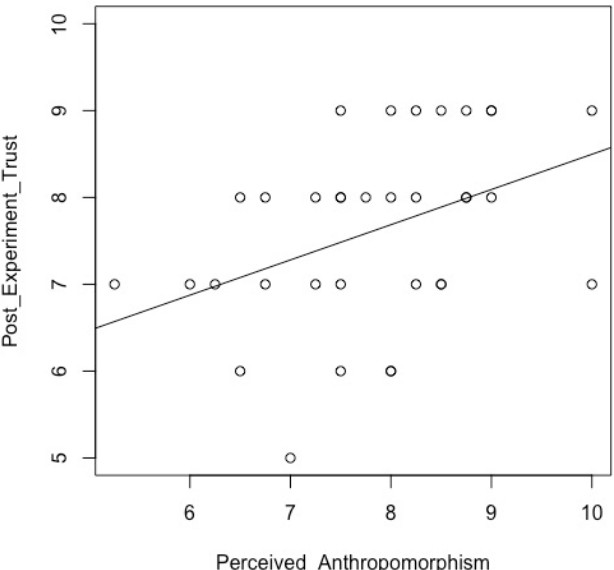

**Figure 9.** Correlation ($\rho$ = 0.45, *p*-value < 0.05) between perceived anthropomorphism (0: Not at all anthropomorphic—10 Very anthropomorphic, according to [14]) and post-experimental trust (according to the scale adapted from [37] (0: Not trustful—10 Very trustful).

### *3.4. Performance*

The fourth question (Q4; see Section 1) was addressed by the hypothesis that, during TOR 60 and TOR 10 scenarios, driving performance (reaction time) is better when the takeover request is emitted by the visual assistant than by the voice assistant or the baseline interface.

Participant reactions were measured using SCANeR Studio data (reaction time, deviation from the axis, and time to collision) aggregated as a performance indicator. Analyses validated this hypothesis for the TOR 60 scenario.

### 3.4.1. TOR 60 Scenario

Here, participants received the takeover request alert 60 s before the end of the automated driving mode. In the voice and visual assistant conditions, the vocal message explained the cause of the takeover request (see Appendix A). This explanation was not available in the Baseline condition. Median participant reaction time (11.3 s) in the visual assistant condition was significantly shorter than in the vocal assistant or baseline conditions (Figure 10) as shown by the Kruskal–Wallis test (Chi-squared = 7.9595, df = 2, *p*-value = 0.0186).

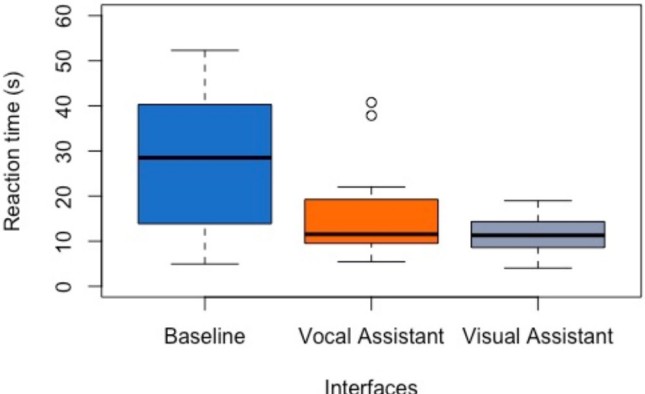

**Figure 10.** Reaction time in TOR 60.

Thus, participants respond more quickly to the takeover request when it was emitted by the visual assistant compared to the baseline condition. These results were confirmed using a pairwise Mann–Whitney test with a Bonferroni correction (Table 9).

**Table 9.** Performance TOR 60, the * sign indicates when the statistical test results are significant.

|  | **Visual Assistant** | **Vocal Assistant** |
|---|---|---|
| Vocal Assistant | 0.7706 |  |
| Baseline | 0.0068 * | 0.0895 |

3.4.2. TOR 10 Scenario

In this scenario (see Figure 11), participants received a takeover request alert 10 s before the end of the automated driving mode. In the vocal assistant and visual assistant conditions, the assistant's voice message explains the cause of the takeover request (see Appendix A). This explanation is not available in the baseline condition. The hypothesis was that the assistant's explanation helped the participant better understand the situation and take over control more quickly in the voice and visual assistant conditions. Data did not support this hypothesis. Instead, reaction times were long (+7 s) in all groups and with no significant differences as shown by the Kruskal–Wallis test (chi-squared = 0.8631, df = 2, $p$-value = 0.6495).

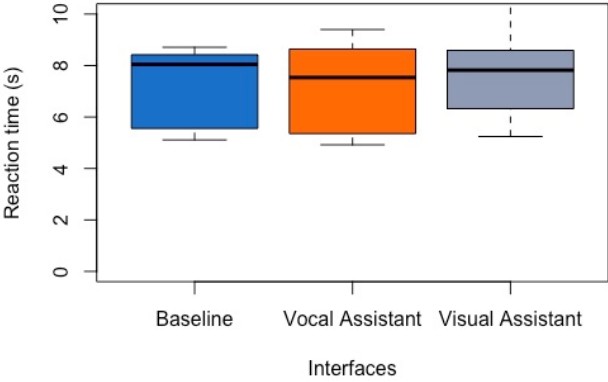

**Figure 11.** Reaction time TOR 10.

## 4. Conclusions

### 4.1. Anthropomorphism

Anthropomorphic attributes such as voice or visual embodiment (voice and visual assistant conditions) added to a non-anthropomorphic interface (baseline condition) did not increase perceived anthropomorphism of the system, as was first hypothesized. Surprisingly, the three interfaces all received high anthropomorphism scores (around 7 on a scale of 10). Several explanations can be emphasized: the anthropomorphic attributes may not have reached the required quality to affect participants' perception. Indeed, improvements could be made to both the voice (tone variations) and the visual (animation) of the assistant. Alternatively, perhaps measured anthropomorphism is not influenced by the three interface types. The literature identifies two types of anthropomorphism [46,47]: conscious (mindful) and unconscious (mindless). The former is the ability to perceive human qualities in a machine, including its "intelligence" and the latter an individual's ability to view a machine as a "living entity". Our measurement did not distinguish these two types. More research could show whether anthropomorphic attributes might impact only one of these.

### 4.2. Trust

The hypothesis that higher perceived anthropomorphism led to higher trust was partially supported. Two scales measured trust (global and multidimensional) at the beginning

of the experiment and at the end of each driving session. There was a moderate yet significant correlation between global trust and perceived anthropomorphism across participants regardless of the interface. This supports our hypothesis that perceived anthropomorphism and trust are positively correlated.

However, when comparing the three interface conditions across the experimental sessions, the pattern became more complex. Measurements revealed an initial difference in trust between the baseline and the two assistant conditions: the initial global trust of participants in the baseline group was significantly higher than that of the voice or visual assistant groups. There was no initial difference in the multidimensional trust score. At the end of the training session, both global and multidimensional trust were significantly higher in the Baseline group. All the metrics converged during the experimental trials when participants tested the interfaces in realistic conditions. At this stage, there were no significant differences among the three conditions regardless of the questionnaire used to assess trust.

We observed that the global trust level is the same for our three interfaces, baseline, voice, and visual assistant. According to our results, anthropomorphic interfaces cannot be recommended to car makers as a lever for increasing users' trust in driving automation. However, further research is needed to fully address this statement. Future research on this topic, may include an improved version of the virtual assistant, a larger sample group, and other factors contributing to users' trust such as acquired experience (driving scenarios participants are exposed to), and the system's level of transparency. Another focus of research could be how to best calibrate trust rather than simply increasing its level.

Another topic of interest is the discrepancy between the two trust assessment methods in which global trust scores seem more variable than multidimensional ones, especially for initial trust. Our results suggest that measuring trust with only one question could be less reliable than using a combination of questions and could lead to erroneous conclusions. Furthermore, since results obtained using the two methods seem to converge after the training session, it seems that participant's adaptation time and adjustment of trust levels should be considered when preparing the experimental protocol. Understanding this adaptation phenomenon could lead to further research in this work.

The hypothesis that higher anthropomorphism would positively impact driving performance was partially confirmed by the study of reaction time after a handover event in two scenarios (TOR 60 and TOR 10). The reaction time was significantly shorter in the visual assistant condition for a takeover request in 60 s. (TOR 60). For a 10 s takeover request (TOR 10), there was no significant difference in driving performance among the different interfaces. In that scenario, the addition of longer voice messages (vocal assistant) or of a hologram (visual assistant) did not affect reaction time.

This research partially supported our four hypotheses. There was a correlation between perceived anthropomorphism and trust. The anthropomorphic assistant, however, cannot be considered a better choice for improving these measures. All three interfaces, including the Baseline, received a high anthropomorphism score and, although initial trust was slightly higher in the baseline group, all groups reached the same level of trust after the training phase. As predicted, the visual and vocal assistant interfaces resulted in a faster takeover response but only in the 60 s takeover scenario and not in the more urgent 10 s one. These results are encouraging and raise exciting questions that require future research.

**Author Contributions:** All authors participated in the conceptualization, principally C.L.-G.; methodology: C.L.-G. and V.R. supervised by N.L., J.-M.A., B.L. and K.A.-F.; investigation and validation were performed by C.L.-G. supervised by N.L., J.-M.A. and K.A.-F.; formal analysis was performed by C.L.-G.; writing—original draft preparation was performed by K.A.-F. writing—review and editing, K.A.-F., N.L., C.L.-G., V.R. and J.-M.A., supervision was performed by K.A.-F., N.L. and J.-M.A.; funding acquisition was performed by IRT SystemX. All authors have read and agreed to the published version of the manuscript.

**Funding:** This work has been supported by the French government under the "France 2030" program, as part of the SystemX Technological Research Institute within the CMI project.

**Data Availability Statement:** Due to local privacy and ethical policy implementation, the results of this study can not be shared with the public.

**Acknowledgments:** This project is a collaborative project between manufacturers (Renault, Valeo, Saint-Gobain and Arkamys) and academic laboratories IMS-ENSC (Bordeaux) and CeRCA (Poitiers). Its objective is to explore human factors in the context of driving automation. It aims to define a multisensory interface (visual, auditory, and haptic) to accompany drivers in the specific use cases of transfer of control (gradient transition) and reassurance. We would like to thank Sabine LANGLOIS, Jean-Marc TISSOT, and Noé MONSAINGEON for their collaboration in this project.

**Conflicts of Interest:** The authors declare no conflict of interest. The sponsors had no role in the design, execution, interpretation, or writing of the study.

## Appendix A

Questionnaires used in the study.

In this experiment, participants took questionnaires in French. "Initial Trust" was kept in its original form in French and "situational trust", "perceived anthropomorphism" and "global trust" questionnaires were translated from their original form in English to French. Here, we present every questionnaire in English.

I—Initial Trust (adapted from [4]; English version)

Items 6 to 24 from [4]

Scale: Not at all (0) --------------------- Completely (10)

1—Do you think you know the automated car?

2—How much trust would you have in the automated car?

3—Do you think the automated car is useful?

4—Do you think the automated car is necessary?

5—Do you think that the automated car would interfere with your usual driving?

6—In everyday life, you tend to take risks

7—You tend to trust people

8—You believe that trusting someone means being able to trust them with something to do

9—You believe that it is necessary to know a person well to trust him/her

10—You think that the decision to trust someone depends on how you interact with them

11—You are generally suspicious of new technologies (cell phones, computers, internet, microwave ovens, etc.)

12—You think that new technologies are interesting

13—You think that new technologies are dangerous

What risks would you associate with using an automated car?

14—The risk of driving more dangerously

15—The risk of not knowing how to use the system

16—The risk of an accident with the car ahead

17—The risk of an accident with the car behind

18—The risk of being dependent on the system

19—The risk of losing the pleasure of driving

What benefits would you get from using an automated car?

20—Less stressful driving

21—Lightening the driving task (the automated car would allow me to perform a secondary activity such as reading or using my phone)

22—Easier driving task

23—Improved driving comfort

24—Safer driving

II—Situational Trust (adapted from [19])

Items 1 to 6 from [19]

Scale: Not at all (0) --------------------- Completely (10)

1—I trust the automation in this situation

2—I would have performed better than the automated vehicle in this situation

3—In this situation, the automated vehicle performs well enough for me to engage in other activities (such as using my smartphone) *

4—The situation was risky

5—The automated vehicle made an unsafe judgement in this situation

6—The automated vehicle reacted appropriately to the environment

7—In this situation, the behavior of the automated vehicle (actions and decisions) surprised me **.

* The NDRT was updated to match the game used in our experiment. In the original questionnaire the task is "reading".

** Item 7 was added by experimenters to include the dimension of surprise in trust evaluation. Here, we present the translated version from French to English.

III—Perceived Anthropomorphism—([7])

Scale: Not at all (0) ---------------------- Completely (10)

1—How smart does this car seem?

2—How well do you think this car could perceive what is happening around it?

3—How well do you think this car could anticipate what is about to happen, before it actually happens?

4—How well do you think this car could plan the best route?

IV—Global Trust (adapted from [18])

Scale: Not at all (0) ---------------------- Completely (10)

In general, do you trust the automated car? *

* The original version of this question mentioned adaptive cruise control. It was replaced by automated car in this experiment.

**Appendix B**

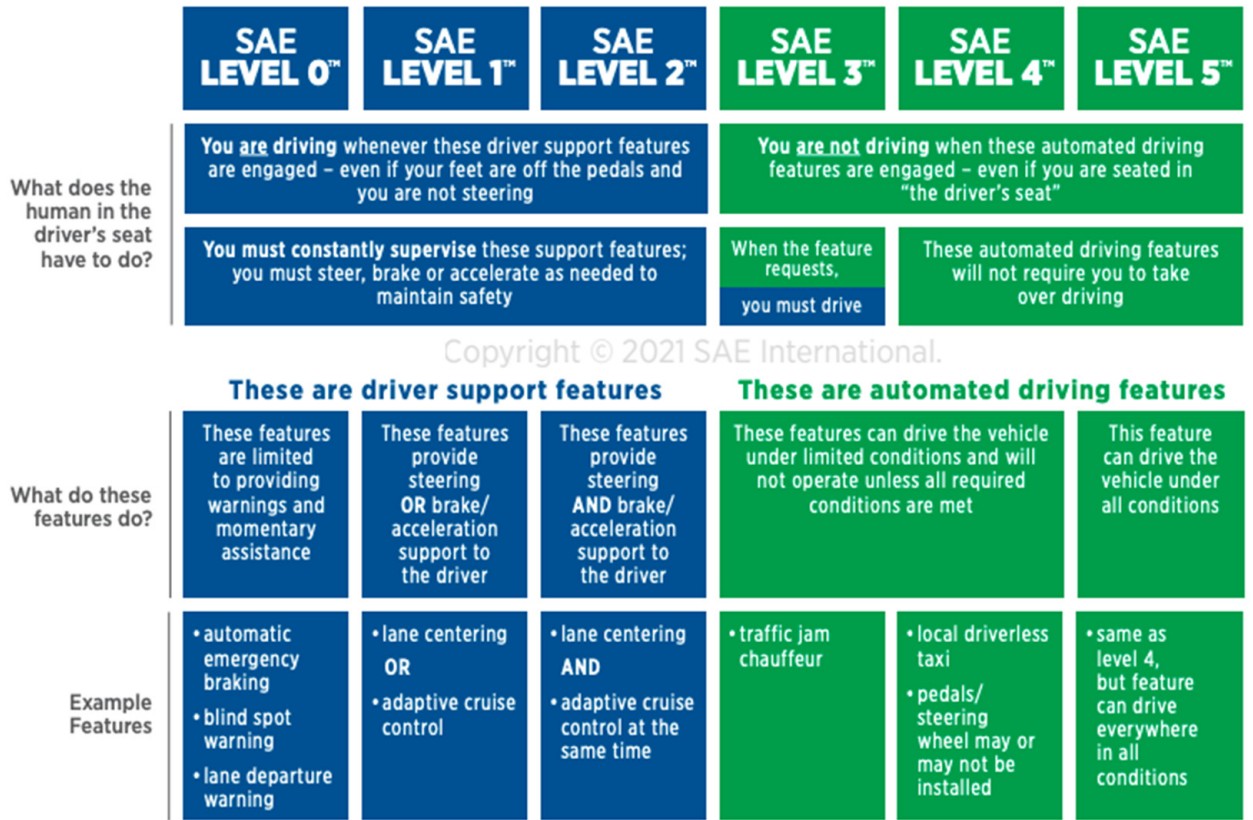

**Figure A1.** Description of different levels of car automation by SAE International.

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
