# Peer review of "Anthropomorphic Design and Self-Reported Behavioral Trust: The Case of a Virtual Assistant in a Highly Automated Car"

_machines, doi:10.3390/machines11121087_

Round 1

Reviewer 1 Report

Comments and Suggestions for Authors

The subject of this research is interesting as the authors try to study the relation between the levels of trust that the humans are showing to the vehicle they are in and the perceived anthropomorphism that its autonomous driving systems are exhibiting. Indeed, the authors submitted a reassessed version of their initial contribution. Some improvements have been made but should be complemented with further referencing material to background works, while a more elaborated research description should be necessary, as they provide marginal corrections to the reviewer suggestions. They also chose to exclude parts that were not properly explained in their former submission instead of providing a satisfactory rationale for their presence, i.e., to use them to reinforce the questioning material and/or the motives of their work.

Indicative issues:     

The authors should explicitly declare the type of their contribution (i.e., article at the top left of the manuscript)

Fog scenario was an interesting testing case to be studied that has been removed from the previous article version.   

Line 109: Reference should be corrected.

Figure and Table references inside the manuscript don’t need to be in bold.

Figures 6, 7 to be magnified in order the text describing blocks to be more readable.

Spacing irregularities should be fixed in section titles.

Supplementary material might be incorporated into the main article as appendices.

Comments on the Quality of English Language

Reviewer suggestions should be better covered. Further reassessment is required and formatting issues to be fixed. 

Author Response

Thank you for reviewing our work.
Many thanks for your pertinent remarks, comments and questions.

Related to the question:

Fog scenario was an interesting testing case to be studied that has been removed from the previous article version?

Response:

The presence of the Fog scenario in the first version of this paper is an editing mistake on our part. The purpose of this scenario is different from TOR 60 and TOR 10 where we look at the impact of anthropomorphism on participants’ takeover performance and trust levels. Indeed, Fog scenario has been designed to assess how participants perceive the vehicle behavior in a safety related scenario. Due to this difference, we decided to present the results from this scenario in a separate publication. 

The answers to others questions are in the blue colore in the paper.

Form-related comments are also taken into account directly in the paper.

Best regards

Reviewer 2 Report

Comments and Suggestions for Authors

This paper shows the study of behavioral trust, the case of a virtual assistant in a highly automated car. This topic is very interesting, and it would be an important factor of deciding the direction of feature development in the virtual assistant system. The reviewer has the following comments:

-          Some editing errors on line 335 and line 336.

-          The reaction times of TOR 10 and TOR 60, if there is no major time difference in TOR10, maybe the Baseline can be implemented to save the cost of cars.

-           It would be great if authors can discuss more about how to gain driver’s trust through the virtual assistant system. 

Author Response

Thank you for reviewing our work.
Many thanks for your pertinent remarks, comments and questions.

The answers to questions are in the blue colore in the paper.

Form-related comments are also taken into account directly in the paper.

Best regards 

Reviewer 3 Report

Comments and Suggestions for Authors

See details in the attached document.

Author Response

Thank you for accepting to review our paper.

Many thanks for your comments suggestions and questions.

Question:

The correlation between anthropomorphic perception and post-experiment trust levels may be better represented using a non-linear correlation. 

Response

Our goal with this analysis was to evidence a connection between anthropomorphism and trust. This simple model (post-test trust predicted by perceived anthropomorphism) showed relatively conclusive results, with a modest yet significant correlation. We could have tried to fit a curve in our dataset in order to reach a more optimal model. It is not clear from our dataset that this would be the case. More importantly, we do not have a theory that would justify such a non-linear relationship. More specific research is needed to address this question.

Responses to other questions and comments are in blue color in the paper

Form-related comments are also taken into account directly in the paper

Best regards

Round 2

Reviewer 1 Report

Comments and Suggestions for Authors

The authors incorporated most of the proposed suggestions and delivered a satisfactory outcome. Reviewer feels covered.

- At the begining, they should completely omit "Type of the Paper" and just write "Article".  

- I suppose they mean "purple" instead of "blue" in their response for marking corrections.  

Author Response

Thank you for your commentes

In fact, the blue color chosen is different between the first and second parts. We unified and chose a single blue

We also corrected the paper type

Reviewer 3 Report

Comments and Suggestions for Authors

This study explores the influence of anthropomorphic design on user trust in the context of highly automated cars. The research involves a virtual assistant with two design levels: "voice-only" and "voice with visual appearance" (a three-dimensional holographic model integrated into the driving simulator). Through a driving simulator study, three interfaces were compared, including the two virtual assistant versions and a baseline interface with no anthropomorphic attributes. Trust versus perceived anthropomorphism was measured, and the evolution of trust across various driving scenarios was examined. The study also analyzed participants' reaction time to takeover request events. The findings revealed a significant correlation between perceived anthropomorphism and trust. However, the tested interfaces did not significantly differ in perceived anthropomorphism, and trust converged over time. Additionally, the anthropomorphic assistant positively impacted reaction time for one takeover request scenario. The study concludes by discussing methodological issues and implications for design and suggesting directions for further research.

1.      introduction

l  To enhance the description of trust, it is recommended to incorporate additional insights into relevant measurement methods.

2.2 Experimantal factors

l  The paragraph contains irrelevant and extraneous text (Error! Reference source not found.). It is advisable to remove this text.

2.3 Participants

l  The experimental group consists of only 12 participants, resulting in a limited dataset, which may contribute to less pronounced differences in the results.

2.6 Data collection and analysis

l  Figures six and seven should be elaborately explained to facilitate a better understanding for the readers.

3.3 Correlation between anthropomorphism and trust

l  The correlation between anthropomorphic perception and post-experiment trust levels may be better represented using a non-linear correlation.

Author Response

Thank you for accepting to review our paper.

Many thanks for your comments suggestions and questions.

Below are our answers 

Question:

The correlation between anthropomorphic perception and post-experiment trust levels may be better represented using a non-linear correlation. 

Response

Our goal with this analysis was to evidence a connection between anthropomorphism and trust. This simple model (post-test trust predicted by perceived anthropomorphism) showed relatively conclusive results, with a modest yet significant correlation. We could have tried to fit a curve in our dataset in order to reach a more optimal model. It is not clear from our dataset that this would be the case. More importantly, we do not have a theory that would justify such a non-linear relationship. More specific research is needed to address this question.

Responses to other questions and comments are in blue color in the paper

Form-related comments are also taken into account directly in the paper

Best regards

Round 3

Reviewer 3 Report

Comments and Suggestions for Authors

This study explores the influence of anthropomorphic design on user trust in the context of highly automated cars. The research involves a virtual assistant with two design levels: "voice-only" and "voice with visual appearance" (a three-dimensional holographic model integrated into the driving simulator). Through a driving simulator study, three interfaces were compared, including the two virtual assistant versions and a baseline interface with no anthropomorphic attributes. Trust versus perceived anthropomorphism was measured, and the evolution of trust across various driving scenarios was examined. The study also analyzed participants' reaction time to takeover request events. The findings revealed a significant correlation between perceived anthropomorphism and trust. However, the tested interfaces did not significantly differ in perceived anthropomorphism, and trust converged over time. Additionally, the anthropomorphic assistant positively impacted reaction time for one takeover request scenario. The study concludes by discussing methodological issues and implications for design and suggesting directions for further research.

Through a meticulous review of the newly submitted version, I posit that the author has made accurate and pertinent modifications. Consequently, I deem the article acceptable for publication.